# Highly Sensitive and Ultra-Responsive Humidity Sensors Based on Graphene Oxide Active Layers and High Surface Area Laser-Induced Graphene Electrodes

**DOI:** 10.3390/nano12152684

**Published:** 2022-08-04

**Authors:** George Paterakis, Eoghan Vaughan, Dinesh R. Gawade, Richard Murray, George Gorgolis, Stefanos Matsalis, George Anagnostopoulos, John L. Buckley, Brendan O’Flynn, Aidan J. Quinn, Daniela Iacopino, Costas Galiotis

**Affiliations:** 1Institute of Chemical Engineering Sciences, Foundation for Research and Technology-Hellas (FORTH/ICE-HT), 265 04 Patras, Greece; 2Department of Chemical Engineering, University of Patras, 265 04 Patras, Greece; 3Tyndall National Institute, University College Cork, Dyke Parade, T12 R5CP Cork, Ireland

**Keywords:** LIG, graphene oxide, humidity, NFC integration, ultrasensitive

## Abstract

Ultra-sensitive and responsive humidity sensors were fabricated by deposition of graphene oxide (GO) on laser-induced graphene (LIG) electrodes fabricated by a low-cost visible laser scribing tool. The effects of GO layer thickness and electrode geometry were investigated. Sensors comprising 0.33 mg/mL GO drop-deposited on spiral LIG electrodes exhibited high sensitivity up to 1800 pF/% RH at 22 °C, which is higher than previously reported LIG/GO sensors. The high performance was ascribed to the high density of the hydroxyl groups of GO, promoted by post-synthesis sonication treatment, resulting in high water physisorption rates. As a result, the sensors also displayed good stability and short response/recovery times across a wide tested range of 0–97% RH. The fabricated sensors were benchmarked against commercial humidity sensors and displayed comparable performance and stability. Finally, the sensors were integrated with a near-field communication tag to function as a wireless, battery-less humidity sensor platform for easy read-out of environmental humidity values using smartphones.

## 1. Introduction

Humidity sensors play an important role in many industrial and day-to-day life applications including environmental monitoring, smart agriculture/home, health, consumer electronics, and the IoT [1,2,3,4]. Up to now, many different transduction methods have been explored, including capacitance [5], resistance [6], optical fibers [7], field effect transistors [8], and quartz crystal microbalance [9]. Many materials have been explored as humidity sensing layers to develop devices displaying high sensitivity, a good linear range, a fast response, and fast recovery times, such as carbon nanotubes [10], metal oxides [11], polymers [12], and graphene-like materials [13]. Among graphene-like materials, graphene oxide (GO) has been one of the most investigated materials. GO exhibits excellent humidity sensing capabilities [14,15,16,17]. The uniqueness of GO as a humidity sensing layer is related to its morphology and surface chemistry. The surface is characterized by basal planes and edges rich in oxygen functional groups (including hydroxyl, epoxy, and carboxylic acid groups) [18], which enhance GO’s hydrophilic properties, responsible for its sensitivity to water molecules. Exploiting these properties, Bi et al. fabricated ultrahigh sensitivity and fast response time humidity sensors in the range of 15–95% RH. The sensors were obtained by drop-casting GO on interdigitated electrodes and exhibited a capacitive response up to 37,800%, >10-times higher than that of conventional sensors [19]. Zhang et al. deposited a graphene oxide/poly (diallyldimethylammonium chloride) (GO/PDDA) composite film on spiral electrodes by a layer-by-layer technique and obtained an unprecedented response of up to 265,640% in the range of 11–97% RH. The associated ultrafast response and recovery times were exploited to demonstrate human breath monitoring capabilities [20]. Other ultrafast capacitive sensors were fabricated by spin coating thin layers of GO onto Al interdigitated electrodes [21]. More recently, in an attempt to reduce the fabrication cost of devices and to increase the versatility of use, novel ultrasensitive devices were realized by the deposition of thin GO layers on Ag electrodes fabricated by screen printing [22] or inkjet printing [23]. In general, during the design of humidity sensors, particular attention is dedicated to the sensing layer and its performance in terms of sensitivity and response time. However, as demonstrated by the latter examples, the incorporation of low-cost printing methods for electrode fabrication enables the creation of lower-cost devices. Additionally, these methods can incorporate flexible polymer substrates. This in turn enables the generation of flexible devices with more versatile applications and IoT integration.

In recent years, direct laser writing methods have been extensively used for the fabrication of graphene-like materials on flexible polymer substrates [24,25]. In this approach, laser irradiation (most commonly, high-power CO_2_ lasers) of polyimide films leads to the formation of an electrically conductive laser-induced graphene (LIG) material [26,27,28]. Patterning of the desired LIG structures is easily performed on a range of flexible substrates, without the use of chemicals or masks and at ambient temperature/atmospheric conditions. Direct laser writing methods are particularly suitable for the fabrication of low-cost sensing platforms. The LIG formation is ascribed to the carbonization of polyimide in the presence of atmospheric oxygen, leading to thermal conversion of sp^3^ carbon atoms into sp^2^ carbon atoms and the formation of conductive and porous structures [24,25]. The LIG morphology is characterized by a high surface area, high porosity, and a high density of edge planes. These qualities make LIG particularly suitable to sensing and energy storage applications. Since its discovery in 2014, LIG structures have been extensively used as electrodes for energy storage applications [29,30] and for on-chip electrochemical sensors [31] and biosensors [32] for environmental [33,34] and human health applications [35]. Recently, Zhu et al. combined GO and interdigitated LIG electrodes for humidity sensing. The range of 11–97% RH was investigated. The effects of GO layer thickness and electrode spacing were explored, and monitoring of breathing and non-contact fingertip proximity was demonstrated [36]. In another recent example, Kulyk et al. used LIG produced by UV irradiation of filter paper as a humidity sensor, showing sensitivities up to 1.3 × 10^−3^% RH [37].

In this work, we fabricated ultrasensitive humidity sensors by the combination of spiral LIG electrodes and GO layers. A 450 nm laser was used. The sensors displayed a wide operating range (0–97% RH), excellent stability, and short response/recovery times. Sensor functionality was studied in the temperature range of 19 °C–27 °C. The effect of GO thickness was investigated, as well and the difference between spiral and interdigitated electrode geometries. In its optimized conditions, the sensor exhibited sensitivity of 1800 pF/% RH at 22 °C and response/recovery times shorter than 16 s. The sensor was also integrated into a near-field communication tag to function as a wireless, battery-less humidity sensor that could read environmental humidity values using a smartphone.

## 2. Experimental

### 2.1. Materials

Sodium hydroxide (NaOH), lithium chloride (LiCl), potassium acetate (CH_3_COOK), magnesium chloride (MgCl_2_), potassium carbonate (K_2_CO_3_), magnesium nitrate (Mg(NO_3_)_2_), copper chloride (CuCl_2_), sodium chloride (NaCl), potassium chloride (KCl), and potassium sulphate (K_2_SO_4_) were purchased from SIGMA ALDRICH and used directly without any further modification. Polyimide films with a thickness of 80 μm were purchased from Radionics and used without further treatment. All solutions were prepared using deionized Milli-Q water (DIW, resistivity 18.2 MΩ cm).

### 2.2. Fabrication of Humidity Sensors

Laser-induced graphene electrodes were fabricated by direct laser writing of polyimide films, as previously described [38]. Briefly, polyimide films were “written” with a KKmoon Compact Automatic Desktop Laser Engraving Machine equipped with a laser with 3 W power and an illumination wavelength of 450 nm. A glass slide was used as a rigid substrate to support the polyimide tape, which was irradiated at 30% laser power to obtain the designed spiral electrode structures by laser raster scanning. The obtained structures were washed with acetone, followed by isopropanol and DIW water before use to remove any residues from the laser engraving process. Two spiral-like structures were used as interdigitated electrodes. The overall pattern had a dimension of 150 mm^2^ (see Figure 1a and Appendix A for further details). Copper wires and Ag paints were used to externally connect the sensors.

The humidity sensing layers were created by drop-casting of 150 μL GO aqueous dispersions of different concentrations (0.33 mg/mL, 0.67 mg/mL, 1.33 mg/mL, and 2 mg/mL) on the LIG electrodes. GO was synthesized based on a two-step oxidation process, as previously reported [39], where single and multiples layers of GO were collected in DIW by a combination of ultrasonication and centrifugation steps. After these steps, an aqueous dispersion of GO was prepared at 2 mg/mL, which was further treated by sonication for 6 h and then diluted with the appropriate amount of DIW to form the above dispersions. Before deposition, the electrodes were immersed in DIW for 1 min to remove the air bubbles from their structure. The LIG/GO sensors were dried in a N_2_ atmosphere overnight before testing.

### 2.3. Materials’ Characterization

The morphological characterization of the obtained sensors was performed by a cold-cathode field-emission scanning electron microscope (SEM, JSM-7500F, JEOL U.K., Ltd., Hertfordshire, UK.) operating at a 5 kV acceleration voltage. AFM images of the sensor surface were collected by contact mode (Bruker, Dimension-Icon, MA, USA.). Images were obtained using ScanAsyst-Air probes (silicon tips on silicon nitride cantilever, Bruker) with a 0.4 N m^−1^ nominal spring constant of the cantilever. Raman Spectra were taken with at 514.5 nm (2.41 eV) laser using a MicroRaman (InVia Reflex, Renishaw, Gloucestershire, UK.) setup. The laser power was kept below 1.5 mW on the sample to avoid laser-induced local heating, while an Olympus MPLN100x objective (NA = 0.90) was used to focus the beam on the samples. The XPS measurements were carried out in an ultra-high vacuum system (UHV), which consists of a fast entry specimen assembly, a sample preparation, and an analysis chamber equipped with a dual anode (Al/Mg) X-ray gun and an LH10 electron analyzer. The base pressure in both chambers was 1 × 10^−9^ mbar. The unmonochromatized MgKα line at 1253.6 eV and an analyzer pass energy of 36 eV, giving a full width at half maximum (FWHM) of 0.9 eV for the Au 4f7/2 peak, were used in all XPS measurements. The XPS core level spectra were analyzed using a fitting routine, which can decompose each spectrum into individual mixed Gaussian-Lorentzian peaks after a Shirley background subtraction. The sample was mounted onto a Si substrate with dimensions 1.5 × 1.5 cm^2^. Finally, the capacitance and impedance of the sensors were recorded by a computer-controlled LCR (E4980A, Agilent, CA, USA).

### 2.4. Humidity Sensors’ Characterization

Figure 1 outlines the method for the sensor fabrication and the preparation of the humidified environments. For the impedance and dynamic response measurements of the LIG/GO sensors, 0–97% RH test environments were created using different saturated salt solutions and N_2_ flow, which provided a stable and controlled RH level at their equilibrium states in a closed container (see Figure 1 for more details). The experiments were performed at room temperature. Saturated solutions of NaOH, LiCl, CH_3_COOK, MgCl_2_, K_2_CO_3_, Mg(NO_3_)_2_, CuCl_2_, NaCl, KCl, and K_2_SO_4_ in a closed vessel were used to obtain approximately 7.5%, 11%, 23%, 33%, 43%, 53%, 67%, 75%, 85%, and 97% RH levels, respectively (see Figure 1b). A 0% RH was obtained in a N_2_ flow chamber. The capacitance response and the impedance spectra of the sensors were measured using an Agilent E4980A LCR meter, which was controlled by a PC through an RS-232 interface. The response of the sensor as a function of RH was performed by exposing the sensor to the inside of the closed vessels with the different RH levels for the intake/outtake of water molecules. The figures of merit used for the evaluation of sensor performance were the normalized response (R) and sensitivity (S), determined by R = C/C_0_ = (C_x_ − C_0_)/C_0_ × 100% and S = (C_x_ − C_0_)/(RH_x_ − RH_0_), where C_x_ and C_0_ are the capacitance of the sensor at the x% and 0% RH levels, respectively. For stability and repetitive measurements, a custom-made relative humidity chamber was designed for rapid switching between high and low relative humidity, at a constant temperature of 22 °C. This gas flow system is displayed in Figure 1c. Two separate controlled N_2_ flows were used for the system. The humidified air was created using two bubble systems at 50 °C and 22 °C, respectively, while for the dry air, the N_2_ flow was used. Finally, the different RH levels were achieved by different mixing ratios of the two flows. In addition, a commercial humidity and temperature sensor, AM2302, was used to monitor and record the humidity and temperature in the chamber, which was controlled by an Arduino UNO.

## 3. Results and Discussion

Humidity sensors were fabricated by drop-casting GO aqueous solutions of different concentrations on LIG spiral electrodes fabricated by direct laser writing of polyimide, as illustrated in Figure 1a. The full characterization of the LIG material produced by 450 nm laser irradiation, comprising Raman and XPS spectra, was reported in [38] and confirmed the graphene-like structure of the material formed. The LIG sheet resistance was approximately 16 Ω/sq. The size details of the fabricated LIG spiral and interdigitated geometries are reported in Appendix A. The full characterization of the GO sensor component is presented in Appendix A. SEM and AFM measurements show the small lateral size (<7 μm) of the GO flakes, while Raman and XPS show the presence of a large number of defects in the lattice [40,41,42], with an O/C ratio of 0.6. In addition, compared to the previous work [39], a remarkable increase of the hydroxyl XPS peak (287 eV) was observed, as a result of GO post-synthesis sonication treatment. The humidity sensing performance (impedance and dynamic response measurements) was characterized by exposure of the sensors to saturated environments of different RH (see Figure 1b); stability and repetitive measurements were performed with the use of a custom-made relative humidity chamber, as illustrated in Figure 1c.

Figure 1a shows an SEM image of a bare LIG electrode, displaying the characteristic high surface area and porous morphology of the material. The high magnification SEM image (see Figure 1a, inset) displays the high density of defects and edge planes, as already reported for LIG materials obtained by visible laser irradiation of polyimide sheets [38]. Upon deposition of the GO sensing layer (0.33 mg/mL of GO dispersion), a uniform and continuous coverage of the electrode was achieved, as shown in Figure 1b. The thickness of the GO film was estimated to be at 50 nm from the AFM measurements. The high magnification image of Figure 1c displays a large density of folds and wrinkles over the entire sensing area (LIG and also polyimide surface between the electrodes), suggesting that the observed morphology was mainly due to the crumpling of thin GO sheets rather than produced by the defective nature of the LIG underlayer. The cross-section SEM image of Figure 1d clearly shows the differentiation between the polyimide (PI), LIG, and GO layers.

Figure 2 shows the characterization of the GO/LIG humidity sensor’s performance. Figure 2a shows the capacitance response across the range of applied frequencies of 20–100 kHz measured with an AC voltage of 500 mV. An increase of capacitance with RH was observed for all investigated frequencies. The rise in capacitance was sharp at low frequencies and decreased in magnitude with increasing frequency. This behavior is in agreement with reported literature data for similar systems [19,20] and was strongly related to the enhancement of dielectric constant and polarization effects upon water adsorption on the GO sensing layer. In the initial state (0% RH), the capacitance did not show a strong frequency dependence. However, as the RH increased, more water molecules became physiosorbed by the oxygen functional groups between the GO layers, leading to the enhancement of leak conductivity γ [43]. In addition, the relative dielectric constant increased, due to the presence of free water atoms at high RH levels. The fabricated sensor displayed a capacitance that followed the equation [44]: $C=\left(\varepsilon_{r}-i\gamma /\omega \varepsilon_{0}\right)C_{0}$, where $\varepsilon_{r}$ is the relative dielectric constant, $\varepsilon_{0}$ is the permittivity of a vacuum, and ω is the frequency. According to this equation, maximum capacitance values are reached at low frequencies and high RH levels. At high frequencies, the capacitance decreased due to the inability of water molecules to tune their polarization to the alternating electric field.

The occurrence of these processes was also monitored by impedance spectroscopy. Figure 2b shows the impedance spectra measured in the frequency range of 20–100 kHz for all RH levels. In order to allow better comparison on the same scale, the spectra above 11% RH were multiplied by different factors to compensate for the decrease of both the imaginary and real part of impedance with the increase of RH, due to their inverse relation with capacitance and conductivity [45]. At low RH values (7–53%), the spectra displayed a large semicircle and only a small line at low frequencies. With the increase of RH, the semicircle decreased in size until it became suppressed for RH higher than 84%. The linear behavior, which was barely observable at low RH, became the dominant feature at high RH. The semicircle resulted mainly from the intrinsic impedance of the sensing film [43,46]. It reflected the high resistance of the bare GO layer and its low conductivity at low levels of physiosorbed water (low RH). The progressive decrease of the semicircle in the impendence spectra was associated with the increase in electrical conductivity. This is related to the formation of higher levels of physiosorbed water layers and the associated polarization and diffusion processes occurring in the sensing layer. It has been speculated that the straight line might result from the ionic and/or electrolytic conductivity, due to the formation of H_3_O^+^ at high RH values [19,46,47].

The operating frequency of 500 Hz was chosen to perform further sensor characterization. At lower frequencies, the capacitance showed instability at low RH levels, while at higher frequencies, the water molecules could not be tuned. At frequencies close to 500 Hz, all sensors, with different concentrations of GO, showed more stable capacitance characteristics for all the range of RH values. Figure 2c shows the dynamic capacitance response of the GO sensor at 500 Hz. The switching RH test was performed by exposing the sensor to various humidity environments between 0 and 97% RH. Each exposure was 90 s long. Between exposures to different environments, the sensor was exposed to 0% RH for 90 s. The graph showed a clear increase in the dynamic response with the increase of RH, associated with the adsorption of water molecules on GO. This behavior was in line with what has been reported for GO-based humidity sensors. According to the literature, different water physisorption processes dominate for different water concentrations [19]. At low water concentrations (low RH values), water molecules were physiosorbed on the GO layer through strong double-hydrogen bonding. This greatly reduced mobility, therefore keeping conductance and capacitance low. As RH increased, the water molecules physiosorbed into multiple layers and were ionized to produce H_3_O^+^. The molecules became more mobile and ultimately exhibited liquid-like bulk behavior, which led to increased conductance and capacitance. Furthermore, at high RH values, water likely penetrated into the GO film, causing hydrolysis of its functional groups and contributing to the observed spike in capacitance.

The sensor displayed a good capacitance response even at low RH values. From 0 to 11% RH, the capacitance rose from 9.5 pF to 14.8 pF. The high initial resistance of the GO film was responsible for the sensitivity observed at low RH. Water molecules strongly physiosorbed to GO through hydrogen bonding. A physiosorbed layer of water formed on the surface. This caused a dramatic change in the capacitance of the sensing layer.

Figure 2d shows the response and recovery times of the sensor at 80% RH. These measurements were performed by using the gas flow system, which provided rapid RH changes without equilibrium requirements. The response time was considered to be the time interval from the last value at 0% RH until the capacitance reached 90% of the maximum value at 80% RH (C_max_ × 0.9). The recovery time was the time interval from the last value at 80% RH until the capacitance reached 90% of the minimum value at 0% RH (C_min_ × 1.1). The recorded response time was 16 s. The recovery time was also fast, with the sensor reaching its initial capacitance value in 9 s. Furthermore, the sensor exhibited a maximum hysteresis of 3.03% at 75% RH, which was calculated based on the absorption and desorption characteristics of Appendix A. Finally, Figure 2e shows the repetitive adsorption/desorption response of the sensor at 20, 40, 60, and 80% RH values. The sensor was exposed to an initial 0% RH environment, moved to 20% RH, and then, back again to 0%. This was repeated five times, with the same protocol being repeated for progressively increased RH environments. The response and recovery times for each RH segment were fast, and the capacitance response increased with RH, following the trend shown in Figure 2c. Importantly, the sensor showed high reproducibility, stable cycling, and a wide dynamic range for the entire range of measured RH. Sensor stability over time was further investigated at three constant RH levels of 25, 50, and 75%, as shown in Appendix A,a. In all three cases, the capacitance increased in the liquefied environment and remained constant throughout the measurement time (30 min).

An in-depth characterization of the effect of GO concentration was performed. Figure 3 shows AFM images displaying the effect of the deposition of 150 μL GO dispersions of increasing concentrations on glass substrates, with an area similar to that of LIG electrodes. The resulting films displayed different morphologies, with the density of wrinkles decreasing and film thickness increasing with increasing GO concentration. The thickness of the film formed from the 0.33 mg/mL GO dispersion was 50 nm, which increased to ~120 nm, ~500 nm, and 1 μm for films prepared by using a 0.67 mg/mL, 1.33 mg/mL, and 2 mg/mL GO dispersion, respectively (see Appendix A for more details). The thickness of the wrinkles gradually increased from 10–15 nm for the 0.33 mg/mL GO dispersion to 450 nm for the GO dispersions of 1.33 mg/mL and higher.

Figure 4 shows the effect of sensing layer thickness in the performance of humidity sensors. As already observed for the 0.33 mg/mL GO layer, all sensors showed an increase of capacitance with RH. The sharpest increase was observed at low frequencies and decreased for higher applied frequencies (see also the capacitance vs. RH at different frequencies data reported in Appendix A). Figure 4a–c show the dynamic capacitance response of the GO sensors at 500 Hz. Each graph shows a clear increase of the dynamic response with the increase of RH, associated with the adsorption of water molecules. Furthermore, individual RH intervals across the different sensors showed increased capacitance response with the increase of GO concentration. However, the thickest GO sensing layer also displayed the slowest response and recovery times. The recovery time increased substantially, from 9 s for the thinnest GO layer to 109 s for the thickest layer (Figure 4d–f) (see also Appendix A). This behavior was clearly related to the thickness and the morphology of the GO sensing films [48]. As the GO thickness increased, more water molecules became trapped between the layers at high humidity levels, leading to their slower release in the recovery phase. Especially in the case of the sensor developed using a 2 mg/mL GO dispersion, the recovery appeared to be incomplete, showing a continuously increasing value of the capacitance at 0% RH, over the repeated cycles (Figure 4i). The repetitive absorption/desorption curves showed good cycling performance and a wide dynamic range for all other GO layer thicknesses (Figure 4g,h).

The sensitivity of all sensors (see Appendix A) was calculated from the formula S = (C_x_ − C_0_)/(RH_x_ − RH_0_). Based on the capacitance values (at 500 Hz) observed in Figure 2c and Figure 4a–c, sensitivities equal to 1800, 3560, 5370, and 13,460 pF/% RH at 0–97% RH were calculated for the sensors developed using 0.33, 0.67, 1.33, and 2 mg/mL GO dispersions, respectively. The normalized sensor response, R, was calculated using the formula R = (C_x_ − C_0_)/C_0_ × 100%. As shown in Appendix A, R values of 1824 × 10^3^, 3430 × 10^3^, 3604 × 10^3^, and 11,710 × 10^3^% were calculated for the same RH range, for the sensors developed using 0.33, 0.67, 1.33, and 2 mg/mL GO dispersions, respectively. Both R and S can be considered as figures of merit for the sensors—they express the variation of capacitance in the measured range and its fluctuations in each RH level, respectively.

In order to study the effect of the LIG electrode architecture on sensor performance, we developed another LIG/GO sensor with interdigitated electrodes, as presented in Appendix A. The active area of the new sensor was slightly smaller (225 mm^2^) than the active area of the spiral electrodes (265 mm^2^). The sensing film consisted of a 50 nm GO film in both cases. Figure 5a presents the dynamic capacitance response of the interdigitated LIG/GO sensor at 500 Hz in various humidity environments between 0 and 97% RH (see also Appendix A for more details). From the analysis of the characteristics of the sensor, it is obvious that it presented a similar capacitance response to the optimized spiral LIG/GO sensor (Figure 2c), with sensitivity at 1430 pF/% RH and normalized response at 2847 × 10^3^%. However, the response and recovery time (Figure 5b) of this sensor were lower (50 s and 18 s), indicating that the electrode geometry significantly affected the overall sensor response.

Table 1 presents the comparison of the optimized spiral GO sensor with other GO-based capacitance humidity sensors. The sensing properties of the developed LIG/GO sensors were comparable and, in many cases, superior to those of the sensors reported in the literature. A sensitivity of 1800 pF/%RH was measured at 500 Hz for the thinnest layer of GO of 0.33 mg/mL. However, as discussed, higher sensitivities were recorded at higher GO concentrations, although a reduction in GO thickness was preferred as it led to a significant reduction in the sensor response and recovery times (Appendix A). However, as presented in Figure 2a, the resonant frequency of the measurements can change the capacitance response, and therefore, both normalized response and sensitivity can be changed (see also Appendix A). Lan et al. [49] report that the highest sensitivity of their interdigitated LIG/GO sensors was 3215.25 pF/% RH at 50 Hz; however, at the same frequency, our sensors’ sensitivity was higher at 7709 pF/% RH, while the thickness of the GO layer was 40-times lower. Direct comparison with the interdigitated GO/LIG sensors reported by Zhu et al. [36] would suggest that a lower performance was recorded for our sensors. However, it should be noted that our sensors had a 4-times less GO active layer, a 2-times smaller sensing area, and a 32-times smaller length/gap electrode ratio. In spite of all these factors, the calculated sensitivity for our sensors was only 4.5-times smaller than those fabricated by Zhu et al., while our sensors’ response time was 3-times faster, with smaller hysteresis. In addition to the sensor characteristics, it should be pointed out that our LIG/GO sensors were characterized at 22 °C, whereas Zhu et al. performed their characterization at 27 °C, an environment containing 1.5-times more grams of water per m^3^ of air at 97% RH [50,51,52,53]. Appendix A presents the fitting of the capacitance values versus absolute humidity at 22 °C. Appendix A shows the capacitance response on all the tested RH levels for 22 °C and 27 °C. The capacitance response of the sensor reached the value of 1,280,000 pF at 97% RH at 27 °C, which corresponds to a sensitivity of 13,190 pf/% RH in the range of 0–97% RH, or 14,870 pf/% RH in the range of 11–97% RH. In other words, at the same temperature, our sensor shows 1.6-times higher sensitivity.

Figure 6 shows the comparison in performance between the fabricated LIG/GO sensor (50 nm GO layer) and a commercial sensor (AM2302) reading humidity and temperature, in the temperature range of 19–27 °C. There was very good agreement in the reading of absolute humidity values between the two sensors. Because relative humidity is a temperature-dependent parameter, it is very important to study the behavior of the LIG/GO sensor when the temperature changes. To convert the capacitance values of the sensor into absolute humidity values, the sensor was first calibrated by using the values obtained from Figure 2c, at 22 °C. Both the LIG/GO and the commercial sensor were placed in a vessel of saturated aqueous Mg(NO_3_)_2_ solution. After an equilibrium period, the vessel was placed in the cooling chamber for 2 h, and then, it was placed under the room conditions. It was observed that both sensors instantaneously detected a drop in absolute humidity at the beginning of the cooling process, due to the lower solubility of water in the atmosphere at decreasing temperatures. The absolute humidity response of both sensors began to increase after two hours, signaling the beginning of the heating cycle, where the solubility of water molecules in the atmosphere increases as a function of temperature.

Finally, the integration of the GO/LIG relative humidity sensor with a battery-less near-field communication (NFC) device was performed to demonstrate fast interrogation of the sensor using a smartphone. The developed sensor was connected to a capacitance-to-digital converter (CDC), which was interfaced with a microcontroller unit (MCU) using the Inter-Integrated Circuit (I2C) protocol, as shown in Figure 7a,b. An NFC Type-5-enabled smartphone was used to wirelessly power the humidity sensor, as shown in Figure 7c. The NFC sensor system comprised an NFC loop antenna, with the smartphone providing wireless power to the sensor via inductive coupling. The NFC sensor transponder also included a Radio Frequency to Direct Current (RF-DC) converter within the NFC radio transceiver, which provided a harvested unregulated DC voltage. The harvested DC voltage was then regulated using a low dropout voltage regulator (STLQ015M21R) that provided a regulated 2.1 V DC voltage. The regulated voltage was used to power the CDC, MCU, and NFC radio transceiver. On power-up, the ambient relative humidity was sensed using the developed LIG-electrode-based humidity sensor. The sensor’s change of capacitance was digitized using the CDC. In addition, the MCU first read the digital values from the CDC, then calculated the Relative Humidity (RH). The measured relative humidity data were read using an NFC-enabled smartphone with the help of a read command, as illustrated in Figure 7c.

In the presented design, both voltage and frequency scaling (VFS) techniques were used to minimize the DC power consumption of the developed sensor with the aim to maximize the wireless communication range. To achieve sub-mW DC power consumption, the NFC-enabled humidity sensor was designed with optimal settings: MCU peripheral clock (*f*_CLK_) = 0.524 MHz; I2C core input clock (*f*_I2C_IN_) = 1.028 MHz; MCU core voltage (*V*_CORE_) = 1.2 V; supply voltage (*V*_DD_) of 2.1 V. As a result, the NFC-enabled humidity sensor required a peak DC power of just 900 µW, which is one of the lowest reported in the literature [59]. A Samsung Galaxy S21 smartphone was used to measure the wireless communication range, achieving a maximum of 4.5 cm in free-space.

## 4. Conclusions

In this work, GO/LIG humidity sensors were fabricated by simple drop-casting of thin GO films on laser-written LIG electrodes on polyimide. The sensors’ performance was investigated by exposure to a 0–97% RH range at 22 °C. The sensors displayed high stability and sensitivity, as well as ultrahigh response and recovery times, in line or superior to other GO-based sensors reported in the literature. This was ascribed to the sonication procedure applied to the GO post-synthesis, which reduced the GO flakes’ size and increased the density of defects and the percentage of hydroxyl groups, therefore increasing its water physisorption capability. The influence of GO active layer thickness was investigated, and it was found that while higher thickness increased sensitivity, it also had a negative effect on the response and recovery times, suggesting that a slightly higher GO thickness might offer a good compromise in performance. The influence of electrode shape (spiral vs. interdigitated) was taken into consideration, and it was found that the higher surface area of the spiral design had a positive effect on the sensitivity. Compared to similar GO-based humidity sensors, the GO/LIG platform displayed comparable or superior sensitivities, a wide RH range, stability, and short response and recovery times. Finally, towards practical applications, the sensor was integrated with an NFC tag that enabled humidity readings through a commercial smartphone.

## Data Availability

Not applicable.

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
