# Peer review of "Highly Sensitive and Ultra-Responsive Humidity Sensors Based on Graphene Oxide Active Layers and High Surface Area Laser-Induced Graphene Electrodes"

_nanomaterials, 2022, doi:10.3390/nano12152684_

Round 1
Reviewer 1 Report
The manuscript titled: Highly Sensitive and Ultra-Responsive Humidity Sensors Based on Graphene Oxide Active Layers and High Surface Area Laser-Induced Graphene Electrodes
by George Paterakis is not describing any top-level work in the area of GO/rGO humidity sensing field. Yet, I see the fabrication techniques and the quantitative data of the performance presented reasonably give this work a chance to be published in the journal of Nanomaterials. Hence, in view of the above; I recommend the publication after thoroughly polishing the manuscript language, which seems unscientific on various occasions. Moreover, the following article could be referred to this work:
S W Hong R Park, H Kim S. Lone, S Jeon, Y W Kwon, B Shin.
One-Step Laser Patterned Highly Uniform Reduced Graphene Oxide Thin Films for Circuit-Enabled Tattoo and Flexible Humidity Sensor Application; Sensors, 18, 1857, 2018
Author Response
Reviewer 1
Comments and Suggestions for Authors
The manuscript titled: Highly Sensitive and Ultra-Responsive Humidity Sensors Based on Graphene Oxide Active Layers and High Surface Area Laser-Induced Graphene Electrodes by George Paterakis is not describing any top-level work in the area of GO/rGO humidity sensing field. Yet, I see the fabrication techniques and the quantitative data of the performance presented reasonably give this work a chance to be published in the journal of Nanomaterials. Hence, in view of the above; I recommend the publication after thoroughly polishing the manuscript language, which seems unscientific on various occasions. Moreover, the following article could be referred to this work:
S W Hong R Park, H Kim S. Lone, S Jeon, Y W Kwon, B Shin. One-Step Laser Patterned Highly Uniform Reduced Graphene Oxide Thin Films for Circuit-Enabled Tattoo and Flexible Humidity Sensor Application; Sensors, 18, 1857, 2018
AUTHORS ANSWER: We thank the reviewer for the constructive comments and suggestions. We have revised the language throughout the manuscript and added a new reference (nr 17) in the revised version of the manuscript.

Reviewer 2 Report
The authors present their work on combining graphene oxide functional layer with laser induced graphene (LIG) electrodes. I have a few suggestions/questions:
(1) Conclusion section includes contradicting statement with that found in the abstract: "ultrahigh response and recovery times".
(2) The authors mentioned their sensitivity is higher than previously reported LIG/GO sensors, but the scope of this comparison is very narrow. More importantly, Table 1 appears to show a much higher sensitivity for [35] and [49] compared to this work. Please let me know if I missed something?
(3) The innovation of this work is unclear. The graphene oxide functional material and laser-induced electrode formation method for humidity sensors have been reported before.
(4) The first figure appears in the results section. I suggest having one earlier in the manuscript.
(5) Text size of some parts of figures, such as Fig 1a is too small for reading.
(6) I recommend giving the limit of detection (in %RH) to enable easier comparison with other humidity sensors in general.
Author Response
Reviewer 2
Comments and Suggestions for Authors
The authors present their work on combining graphene oxide functional layer with laser induced graphene (LIG) electrodes. I have a few suggestions/questions:
(1) Conclusion section includes contradicting statement with that found in the abstract: "ultrahigh response and recovery times".
AUTHORS ANSWER: We thank the reviewer for pointing this out. We have now amended the text in the conclusions to be in line with the text of the abstract.
(2) The authors mentioned their sensitivity is higher than previously reported LIG/GO sensors, but the scope of this comparison is very narrow. More importantly, Table 1 appears to show a much higher sensitivity for [35] and [49] compared to this work. Please let me know if I missed something?
AUTHORS ANSWER: We thank the reviewer for the constructive comment and the opportunity to clarify this issue. The LIG/GO sensors described by references [35] ( [36] in the revised version) and [45] ([46] in the revised version) in Table 1 indeed had higher values of sensitivity. However, these values were extracted directly from publications and referred to the best performance of the sensors both at the selected resonance frequency and at the temperature of the measurements. These parameters were different in our case, and as is widely known, they are particularly critical for the response of the capacitance humidity sensors. Compared to [36], at the same temperature of 27 °C, our sensor presents 1.6 times higher sensitivity (14870 pf/% RH), while compared to [46], the sensitivity of our sensor at 50 Hz was at 7709 pF/% RH, for 22 °C, more than double to the one reported in [46]. An in-depth comparison of all the differences in the sensors is presented in SI, while in the revised manuscript, we included an additional paragraph as below, outlining why our sensors showed better performance compared to those reported in literature.
“Lan et al. [46] report that the highest sensitivity of their interdigitated LIG/GO sensors was 3215.25 pF/% RH at 50 Hz, however, at the same frequency, our sensors’ sensitivity was higher at 7709 pF/% RH, while the thickness of the GO layer was 40 times lower. Direct comparison with the interdigitated GO/LIG sensors reported by Zhu et al. [36] would suggest that a lower performance was recorded for our sensors. However, it should be noted that our sensors had 4 times less GO active layer, 2 times smaller sensing area and 32 times smaller length/gap electrode ratio. In spite of all these factors, the calculated sensitivity for our sensors was only 4.5 times smaller than those fabricated by Zhu et al., while our sensors’ response time was 3 times faster. In addition to the sensor characteristics, it should be pointed out that our LIG/GO sensors were characterized at 22 °C, whereas Zhu et al. performed their characterization at 27 °C – an environment containing 1.5 times higher grams of water per m3 of air at 97 % RH [47-49]. Figure S8a presents the fitting of the capacitance values versus absolute humidity at 22 °C. Figure S8b shows the capacitance response on all the tested RH levels for 22 °C and 27 °C. The capacitance response of the sensor reached the value of 1280000 pF at 97% RH at 27 °C, which corresponded to a sensitivity of 13190 pf/% RH in the range of 0-97% RH, or 14870 pf/% RH in the range of 11-97% RH. In other words, at the same temperature our sensor shows 1.6 times higher sensitivity.”
(3) The innovation of this work is unclear. The graphene oxide functional material and laser-induced electrode formation method for humidity sensors have been reported before.
AUTHORS ANSWER: We thank the reviewer for raising this very important point. Indeed, there are other publications on LIG/GO humidity sensors, as mentioned in the previous comment. However, both the LIG electrodes and the GO of this publication were prepared with different methods. GO synthesis and post-treatment led to a remarkable increase of oxygen groups in its lattice, which are responsible for the physisorption of the water molecules, and consequently for the high measured sensitivity of the sensors. The LIG fabrication for this work was done with a low-power (3 W) 450 nm laser, rather than the high power CO2 lasers used in previous publications, which sensibly reduced fabrication costs while maintaining physico-chemical characteristics of resulting LIG materials comparable or superior to the properties of CO2-produced LIG. Moreover, the combinations of these innovations led to realization of LIG/GO humidity sensors of superior sensitivity and reponse than other reported solutions, including the already reported LIG/GO systems (see our response to point 2 above)
(4) The first figure appears in the results section. I suggest having one earlier in the manuscript.
AUTHORS ANSWER: We agree with the reviewer’s comment, scheme 1 has been moved to the experimental section.
(5) Text size of some parts of figures, such as Fig 1a is too small for reading.
AUTHORS ANSWER: text size correction was implemented, as suggested by the reviewer.
(6) I recommend giving the limit of detection (in %RH) to enable easier comparison with other humidity sensors in general.
AUTHORS ANSWER: We thank the review for this observation. We understand that by LOD the reviewer means the error on a measurement of RH (± x%). For such a general comparison with other humidity sensors, we believe that the investigated RH range (%) and Sensitivities (pF/%RH) presented in Table 1 are sufficient. In our opinion, and according to parameters generally quoted in other publications, the measurement error is not a parameter that we saw commonly quoted, and therefore we do not think that it could easily serve for comparison.

Reviewer 3 Report
In this paper, the authors report an ultra sensitive and responsive humidity sensor. The sensor is based on the deposition of graphene oxide on a laser-induced graphene (lig) electrode. The sensors displayed good stability and short response and recovery times in the wide tested range 0-97% RH. Finally, sensors were integrated with a near-field-communication tag to function as wireless, battery-less humidity sensor platform for easy read out of environmental humidity values using commercial smartphones. I believe that publication of the manuscript may be considered only after the following issues have been resolved.
1. What is the physical mechanism of the excellent performance of this work?
2. In Table 1, it is suggested to adjust according to the order of citation.
3. The text information in Figure 7 is not clear, and the author needs to make adjustments.
4. The introduction can be improved. The articles related to the some applications of graphene-like materials should be added such as RSC Adv., 2022, 12(13), 7821-7829; Electrochimica Acta 2020, 330, 135196; ChemElectroChem, 6 (2019), pp. 5642-5650; Sens. Actuators B Chem. 2018, 260, 529–540; ACS Sustain. Chem. Eng. 2015, 3, 1677–1685.
5. In the abstract part, the author needs to explain the relevant physical mechanism.
Author Response
Reviewer 3
Comments and Suggestions for Authors
In this paper, the authors report an ultra sensitive and responsive humidity sensor. The sensor is based on the deposition of graphene oxide on a laser-induced graphene (lig) electrode. The sensors displayed good stability and short response and recovery times in the wide tested range 0-97% RH. Finally, sensors were integrated with a near-field-communication tag to function as wireless, battery-less humidity sensor platform for easy read out of environmental humidity values using commercial smartphones. I believe that publication of the manuscript may be considered only after the following issues have been resolved.
- What is the physical mechanism of the excellent performance of this work?
AUTHORS ANSWER: We thank the revier for raising this very important point. We attributed the remarkable performance of the sensor to a novel post synthesis treatment applied to GO. An extensive sonication treatment step was applied to the as synthesized GO, which led to a decrese in size of formed flakes, formation of high density defects and increase of oxygen groups in its lattice. These factors all led to an increased physisorption of water molecules, and consequently to the high measured sensitivity of the sensors. A comment on this mechanism has been added to the manuscript in the experimental section and in the general discussion.
- In Table 1, it is suggested to adjust according to the order of citation.
AUTHORS ANSWER: We thabk the reviewer for this suggestion. Table 1 has been readjusted based on the order of citations in the manuscript.
- The text information in Figure 7 is not clear, and the author needs to make adjustments.
AUTHORS ANSWER: We thank the reviewer for this observation. We have tried to make the text related to NFC integration more clear.
- The introduction can be improved. The articles related to the some applications of graphene-like materials should be added such as RSC Adv., 2022, 12(13), 7821-7829; Electrochimica Acta 2020, 330, 135196; ChemElectroChem, 6 (2019), pp. 5642-5650; Sens. Actuators B Chem. 2018, 260, 529–540; ACS Sustain. Chem. Eng. 2015, 3, 1677–1685.
AUTHORS ANSWER: We thank the reviewer for the suggestion. We have carefully evaluated the publications suggested by the reviewer. However, as they are mostly related to general properties of GO or graphene with no direct correlation to humidity sensing applications, we have decided not to include them to our introduction.
- In the abstract part, the author needs to explain the relevant physical mechanism.
AUTHORS ANSWER: As importantly suggested by the reviewer, details of the sonication treatment of GO, leading to an increase of hydroxyl groups and therefore higher humidity physisorption rates, has been included in the abstract.

Round 2
Reviewer 2 Report
The authors have addressed all of my comments/concerns except the sensor accuracy (%RH). This is the single most important parameter for humidity sensors (and many other types of sensors), and a parameter always reported by humidity sensor products. Otherwise it is not possible to compare humidity sensors based on different technologies and detection methods (electrical, optical, chemical). I feel it must be reported to facilitate better comparisons with the literature and current products on the market.
Author Response
We thank the reviewer for their suggestion. Of course, hysteresis (or accuracy) is a critical parameter for the humidity sensors. The inclusion of this figure is a valuable update to the manuscript. As you can see in Table 1, hysteresis is referred to in only 4 of the 10 publications. This is what our previous answer was based on. In our case, the sensor exhibits maximum hysteresis of 3.03 % at 75 % of RH, which was calculated based on the absorption and desorption characteristics of Figure S3. The necessary changes have already been made to the text and highlighted. A new column has been included in Table 1 for comparison.
Reviewer 3 Report
There is no good reply to the comments of the article. It is suggested that the article is unacceptable.
Author Response
The position of Reviewer 3 is regrettable. The initial comments of the reviewer were that publication was advised after resolving a list of five issues. We addressed four of these issues in our response, but failed to find relevant ground for including the five extra references suggested. Given that the reviewer now deems the article unacceptable for publication, we are led to believe that it is for lack of inclusion of these references. We stand by our decision to not include these articles. They are not relevant to the work presented. The articles in question are:
- Multi-mode surface plasmon resonance absorber based on dart-type single-layer graphene – a paper concerning the use of graphene as a plasmon resonance absorber.
- Novel hierarchical sea urchin-like Prussian blue@palladium core-shell heterostructures supported on nitrogen-doped reduced graphene oxide: facile synthesis and excellent guanine sensing performance - a paper concerning electrochemical sensing of guanine with Prussian blue/palladium being the sensing/catalytic element.
- Self-supporting electrode composed of SnSe nanosheets, thermally treated protein, and reduced graphene oxide with enhanced pseudocapacitance for advanced sodium ion batteries – a paper concerning electrochemical energy storage.
- Facile synthesis of Ag@Cu2¬O heterogeneous nanocrystals decorated N-doped reduced graphene oxide with enhanced electrocatalytic activity for ultrasensitive detection of H2O2 – a paper concerning electrochemical detection of H2O2.
- One-step fabrication of graphene oxide enhanced magnetic composite gel for highly efficient dye adsorption and catalysis – a paper concerning a graphene oxide PVA composite gel used for removing dyes (methylene blue and methyl violet) from water.
While most of these papers use graphene oxide, their subjects are distant from humidity sensing. We have been thorough in our literature review for this paper. We have referenced publications regarding graphene oxide fabrication methods, and humidity sensing applications.